# Determining Genetic Markers and Seed Compositions Related to High Test Weight in *Glycine max*

**DOI:** 10.3390/plants12162997

**Published:** 2023-08-19

**Authors:** Zachary Shea, William M. Singer, Luciana Rosso, Qijian Song, Bo Zhang

**Affiliations:** 1School of Plant & Environmental Sciences, Virginia Tech, Blacksburg, VA 24061, USA; zachary.shea@usda.gov (Z.S.); wilmsing@vt.edu (W.M.S.); luciana@vt.edu (L.R.); 2USDA-ARS, Beltsville Agricultural Research Center, Beltsville, MD 20705, USA; qijian.song@usda.gov

**Keywords:** soybean, GWAS, genetics, test weight, protein, oil

## Abstract

Test weight, one of the primary indicators of soybean seed quality, is measured as the amount of soybean seeds in kilograms that can fit into one hectoliter. The price that growers receive for their soybean is dependent on test weight. Over the past 50 years, growers have observed a decreasing trend in test weight. Therefore, it is imperative to understand better the relationship between soybean test weight and other traits to enable breeders to select parental lines with high test weights in breeding programs to ensure the grower’s profitability. The objectives of the study were to identify genetic markers associated with high test weight in soybean and to determine the correlation between high test weight and five important seed composition traits (protein, oil, sucrose, raffinose, and stachyose content). Maturity group IV and V germplasms from the USDA soybean germplasm collection were grown in Blacksburg and Warsaw in Virginia from 2019 to 2021 and were measured for all of the above traits. Results show that test weight values ranged from 62–77 kg/hL over the three years. Multiple single-nucleotide polymorphisms (SNPs) significantly associated with high test weight were found on chromosome (Chr.) 15 along with a couple on chromosome 14, and 11 candidate genes were found near these SNPs. Test weight was found to be significantly negatively correlated with oil content, inconsistently correlated with protein content in all environments, and negatively correlated but not significantly with all three sugars except for raffinose in Blacksburg 2019. We concluded that the genes that underlie test weight might be on chromosome 15, and the validated associated SNPs might be used to assist breeding selection of test weight. Breeders should pay special attention to test weight while selecting for high oil content in soybean due to their negative correlation.

## 1. Introduction

### 1.1. Overview of Test Weight in Soybean

Soybean (*Glycine max*) is the most important oilseed crop due to its low-cost production and its diverse uses in feed and food for its excellent nutrient profile and in paints and biofuels for its high oil content [1]. The United States has been one of the world’s top two soybean producers for decades [2]. Seed quality is one of the primary factors that affect the price that growers can receive for their soybean. Multiple aspects can affect seed quality, including test weight, the amount of diseased seeds, and damaged or disfigured seeds [3]. Of these, test weight is the most important indicator of soybean seed quality because soybeans with higher test weight last longer in storage, having higher seed integrity and being less prone to mold [3]. Additionally, every crop has a standard value for test weight, and if the crop’s test weight is below the standard, the grower can receive a pricing penalty [4]. The standard for soybean is at 60 lb/bu, and values below 54 lb/bu can cause a pricing penalty [4]. This is important because farmers have noticed a slight decreasing trend in test weight for soybean since the 1950s. While in the 1950s almost all soybean had a test weight around 58–60 lb/bu (72.5–75 kg/hL), farmers are now seeing more test weight values that are closer to 56 lb/bu (70 kg/hL) [3]. Even though this is still above the 54 lb/bu (67.5 kg/hL) limit, if this trend continues, farmers will start to see soybean test weight values at or below 54 lb/bu and will be more likely to lose profit.

### 1.2. Breeding Efforts to Increase Test Weight

Although test weight is important in soybean, not much work has been conducted looking into the effect of genotypes on test weight in this crop. Some work has found that genotypes showed a significant effect on test weight in maize [5], wheat [6], and oat [7]. Recently, in soybean, genotypes were found to have a significant effect on test weight in most locations but not all depending on the environment [8,9]. These findings indicate that it is possible to improve test weight through breeding. Despite this, little to no work has been conducted regarding how to assist breeding high-test-weight soybean varieties. In order to address the concern of a decreasing trend of test weight in soybean over time, there is an urgent need to identify, validate, and utilize genetic markers associated with high test weight in soybean breeding as a quick and effective approach. Thus, breeders will shorten the period of development and selection of soybean varieties for increased test weight to ensure farmers’ profitability.

### 1.3. Correlation with Other Traits

While test weight is important in most crops, other traits are also important in plant breeding and must be taken into consideration, such as protein, oil, and sugar content in wheat and soybean. Therefore, it is necessary to gain a better understanding of the relationship between test weight and seed compositions. Although some work has been conducted previously determining this relationship, reports have indicated varying degrees of association. In wheat, some studies found that protein content had a significant, positive relationship with test weight [10,11], while others found a significant, negative correlation [12] or no significant relationship at all [13]. Oil content was found to have a significant, positive correlation with test weight in sunflowers [14] and no significant effect on test weight in oats [15]. For soybean, protein and sucrose content were found to have varying degrees of significant relationships with test weight, varying from positive to negative to not significant, while oil content and test weight had a significant, negative correlation in all experiments [8]. Test weight is known to be heavily impacted by the environment, but no research has been conducted in the Mid-Atlantic region, so it is important to determine the relationship between test weight and these traits in Virginia. Additionally, little to no work has been conducted to understand the relationship between test weight and raffinose family oligosaccharides (RFOs). While these two sugars are not as major as sucrose, reduced RFOs are a target when breeding new varieties for animal feed and human food.

## 2. Results

### 2.1. Test Weight

Test weight values showed a normal distribution across environments and years, ranging from 62 to 77 kg/hL with a grand mean of 70.2 kg/hL and an average standard deviation of 1.94 kg/hL. Figure 1 shows the distribution of test weight values for both locations (Figure 1A), Blacksburg (Figure 1B), and Warsaw (Figure 1C). Blacksburg ranged from 64 to 77 kg/hL with an average of 70.3 kg/hL and with a standard deviation of 1.72 kg/hL for all years, while Warsaw ranged from 62 to 77 kg/hL and had an average of 70.1 kg/hL with a standard deviation of 2.14 kg/hL. Both locations had similar averages, but apparently, Warsaw had a wider spread. Throughout all years and in both locations, there was one accession, PI87059, that consistently had high test weight.

### 2.2. Genome-Wide Association Study

All significant SNPs that were focused on for this project were on either Chr. 14 or 15, with five SNPs being significant in more than one environment, three SNPs being significant in one environment, and the other three being just under the significance threshold (α = 4.91) (Table 1). These two chromosomes were focused on because they were the only ones to have SNPs that were found to be significant or just below the significance threshold across environments. There are three exceptions to this: SNP ss71562017 was included because it had the highest LOD score of −log10(p) in W 2020, and no other SNP from W 2020 was found to be significant. SNPs ss715623162 and ss715619843 were included because they were the only significant SNPs in BB 2020 on chromosomes 14 and 15. Lastly, SNP ss715618025 was included because it was the only SNP on chromosome 14 or 15 that was significant in BB 2021. In 2019, all significant SNPs were found on chromosome 15, and significant SNPs were found on this chromosome in all locations (Figure 2). In 2020 and 2021, there were many additional SNPs on other chromosomes that were found to be significant, but they were not found in any other environments and therefore were not included for further analysis (Figure 3 and Figure 4). Warsaw did not have any SNPs above −log10(*p*-value) of 4.9 in 2020 and 2021 but had multiple SNPs that were close to the threshold (Figure 3 and Figure 4). QQ plots show that the data were normally distributed (Figure 2, Figure 3 and Figure 4).

Table 1 shows SNPs that were above the threshold or just below it in more than one location with their corresponding name, chromosome location, position, and −log10(*p*-values). The SNPs ss715619843, ss715623162, and ss715620172 were included because they were the markers with the highest values for −log10(*p*-value) in BB 2020 (for the first two) and W 2020 (for the third), and these locations did not have any SNPs that were found to be significant in any other environment. Additionally, SNP ss715618025 was included because it was the only SNP that was significant in BB 2021 on chromosome 14 or 15.

### 2.3. Candidate Genes

A total of eight candidate genes were found on Chr. 15 and three on Chr. 14 that were located within 10 kbp of the SNPs that were found to be significant (Table 2). Only a few genes are located within 10 kbp of the significant SNPs, but no genes are located within 10 kbp of ss715623250. Most of the genes were expressed in multiple tissues including leaf, flower, pod, and seed. Some were only expressed in specific tissues, such as Glyma.14g030400 being expressed in only flower tissue, Glyma.15g119200 being expressed in pod and seed, and Glyma.15g122800, Glyma.15g125000, and Glyma.15g127900 being expressed in root tissue. The functions of the candidate genes mostly involve RNA and/or protein binding or regulation. The gene Glyma.15g119200 has a different function that involves seed storage, and the gene Glyma.15g127900 has an unknown function.

Table 2 shows the candidate genes near SNPs that were previously identified as significant or just below the threshold and their corresponding functions. Expression data were obtained from Soybase. Detailed expression data can be found in Severin et al. 2010 [16].

### 2.4. Correlation of Test Weight with Seed Composition Traits

The Pearson’s correlation between test weight and protein and oil content for 2019–2021 and between test weight and raffinose, sucrose, and stachyose content for 2019 varied depending on location and year (Table 3). The correlation between test weight and protein was not consistent due to a positive correlation in three environments and a negative correlation in another three environments. In addition, three environments showed significant correlation, namely, BB and W 2019 with negative correlations and W 2020 with a positive correlation. The correlation between test weight and oil was found to be significantly negative in all environments except for BB 2021. The strongest correlation was found in W 2021 at −0.387. The correlation between test weight and all sugars in 2019 was negative, but only raffinose had a significant correlation coefficient of −0.175 in BB 2019.

## 3. Discussion

While test weight has not received as much attention as other traits such as protein and oil content in soybean, it is a crucial trait for farmers’ profitability. This is coupled with its power to affect the pricing of soybean and the decreasing trend that farmers have observed in test weight in the past several decades [3]. In this study, we were able to identify multiple genetic markers significantly associated with test weight through GWAS, identify potential candidate genes for test weight, and provide information regarding the relationship between test weight and important seed compositions.

Many studies have found a significant relationship between genotype and high test weight in wheat, soybean, and other crops [5,6,7,8]. However, no studies have found SNPs associated with test weight. In this study, we were able to find many significant SNPs, especially on chromosomes 14 and 15. Environments including BB 2020, BB 2021, and Combined 2021 had multiple significant SNPs that were found to be associated with test weight but were not present in any other environment. SNPs on Chr. 14 and 15 were found to be either significantly associated or close to being significantly associated with test weight across years and locations. Of these two, Chr. 15 seemed to be more consistent because it had SNPs that were significant or close to being significant in almost all environments, while Chr. 14 primarily had significant SNPs in BB 2020 and 2021, which indicates that Chr. 15 may have genes that control test weight. Furthermore, 11 of these SNPs were found to be within 10 kbp of a gene. Six of these genes had little to no expression in seed tissue, and due to this they are less likely to be related to test weight. In addition, most of these SNPS are located close to genes that encode proteins to bind and regulate RNA and other protein; ss715620221 is near Glyma.15g119200, which codes for a seed storage protein and has high expression in seed tissue. While this does not guarantee anything, ss715620221 could be a promising genetic marker because test weight is related to seed durability. Three other genes had high expression in seed tissue, so these could be promising genetic markers as well. Additionally, SNP ss715620172 is located near a gene with an unknown function, which encourages future studies to determine if Glyma.15g127900 is related to test weight.

While Chr. 15 was found to consistently have significant SNPs or SNPs close to the significance threshold, there were no SNPs that were found to be significant across all locations and years. This could be explained by the impact that the environment has on test weight. It is known that while genotypes affect test weight, the environment can also significantly influence test weight [8]. BB and W have different climates, with BB being cooler and less humid than W. Three environments, Warsaw 2020, Combined 2020, and Warsaw 2021, had no SNPs that were above the threshold of 4.91. While they did have SNPs that were just below the threshold, it is interesting that no SNPs showed significant association in those environments despite all other environments having at least one significant SNP. Additionally, BB 2020, Combined 2021, and BB 2021 had multiple SNPs on other chromosomes that were significant but not consistently found in the other environments. This contrasts with 2019, as all environments in this year only had significant SNPs on chromosome 15. These two findings could largely be explained by the time of harvesting and rainfall. It is important to note that delayed harvesting has been found to negatively affect test weight in wheat and corn [17,18]. Harvesting occurred normally in 2019 but was delayed by a couple of weeks due to the weather in 2020 and 2021. Because delaying harvesting and rainfall can impact test weight [17,18,19], these differences between years most likely resulted from delayed harvesting in 2020 and 2021. Other factors have been found to affect test weight as well. For instance, planting date and sulfur/phosphorus application have been found to impact test weight [20,21,22]. All tests were planted in mid-May, so it is unlikely that planting date had an effect on test weight for this project, and no fertilizers or nutrient supplements were applied to these tests.

Our correlation study found that the correlation between test weight and protein content was inconsistent. In BB 2019 and W 2019, protein was found to be significantly negatively correlated with test weight, but W 2020 showed significant, positive correlation. No significant correlation was found between high test weight and protein content in the other environments. Additionally, sucrose content was found to have no significant correlation with test weight. Other studies have found inconsistent correlation results between either protein or sucrose and test weight [8,11,12,13]. However, one study did find that sucrose could be significantly, positively related to test weight [8], but this study was conducted in Georgia, a different environment from Virginia, which might be the main cause of the different correlation results. No studies have looked at the relationships of raffinose and stachyose with test weight. While we did find a significant, negative correlation between test weight and raffinose in BB 2019, it was found to not be significant in W 2019, and stachyose was found to not be significant in either location. Based on the inconsistent correlation between sucrose and test weight, it is not surprising that we did not find consistent significant correlations for these traits because these compounds are both sugars and similar to sucrose. On the other hand, oil was found to be significantly negatively correlated with test weight in all environments except for BB 2021. This is similar to the study conducted by Liu et al. 2019 that also found a significant, negative relationship between oil and test weight. This consistent negative relationship between oil content and high test weight could partly explain the decreasing trend in test weight that farmers have observed. One of the main traits that soybean breeders select for is higher oil content. Therefore, breeders may need to modify their selection objectives to balance oil content with increased test weight.

In summary, multiple significant SNPs associated with test weight on chromosomes 14 and 15 were identified, which could be used by breeders to quickly select high-test-weight progenies derived from high-test-weight parents in order to increase the overall test weight of a breeding program’s germplasm. Additionally, a consistent negative relationship between high test weight and oil content was found. This information could be useful for breeders because while they make their selections to ensure high oil content, they should also pay attention to test weight to help offset its decreasing trend. Ultimately, incorporating parents that have high test weight into breeding schemes and taking into consideration the relationship between test weight and oil content will enable breeders to develop varieties with increased test weight to avoid harming famers’ profitability in the long run.

## 4. Materials and Methods

### 4.1. Plant Materials

All accessions were planted similarly to Singer et al. 2022 and Wang et al. 2022 [23,24]. Briefly, a total of 390 soybean accessions from a total of 17 countries from the Southern Core Collection that has been maintained at Virginia Tech were grown in Blacksburg (BB) and Warsaw (W), Virginia, for three years from 2019 to 2021. This panel contains soybean accessions that were grouped by maturity groups, and all accessions belonged to either maturity group IV or V at both locations for all years. Each sample was planted in two replications in 3 m two-row plots with 76 cm of row spacing for each plot in BB and 3 m four-row plots with 76 cm of row spacing in W, with Ellis and AG 4404 being used as commercial checks. All plots were checked for flower color and pubescent color, and any plants that did not match the correct color for that plot were removed. All plots were planted in the beginning of May in both locations and all years. In 2019, all plots were harvested around the beginning of October, but in 2020 and 2021, plots were not harvested until later in October due to weather conditions.

### 4.2. Test Weight

All samples were cleaned to remove split seeds, empty seed coats, pods, and sticks so that only intact seeds remained prior to measuring test weight. Sample seeds were also checked for seed coat and seed hilum color, which served as quality controls to remove any contaminant seeds. Blacksburg had 345, 314, and 241 accessions for 2019, 2020, and 2021, respectively. Warsaw had 314, 267, and 275 accessions for 2019, 2020, 2021, respectively. Test weight was determined by using a 2500 AGRI model. For each sample, 414 mL was used to calculate test weight. All samples were measured three times, and the resulting test weight values were averaged to obtain the final test weight value for the sample. For 2019, due to limited seed amount, the two replications had to be combined into a composite sample prior to measuring, but we were able to average two replications for a mean of the test weight of each accession in 2020 and 2021. Test weight values were adjusted to 13% moisture content according to Liu et al. 2019. Lastly, test weight values were converted from lb/bu to kg/hL by multiplying the lb/bu values by a factor of 1.25.

### 4.3. Genome-Wide Association Study (GWAS)

The genotypic data of all accessions screened by SoySNP50K iSelect Beadchip [25] are publicly available at soybase.org [26], with a total of 35,570 SNPs in this germplasm population. A GWAS was performed by first using the TASSEL 5.0 software to construct a kinship matrix, perform principal component analysis, and analyze association with a mixed linear model (MLM) [27]. The MLM was used to include a kinship matrix (K) with population structure (Q) to improve the statistical power using the Q + K approach [28]. A modified Šidák correction (αsid = 1 − (1 − α)(1/m)) for multiple testing was used to determine any significant markers, with the number of effective markers (Meff) being used instead of total number of markers (m). The Meff was determined to be 4191 through the poolr package in R with the Li and Ji method [29]. A modified significant threshold at α = 0.05 was constructed at −log10(P) > 4.91 to determine the significance of single-nucleotide polymorphisms (SNPs). The qqman package was used to construct QQ and Manhattan plots [30]. In order to find candidate genes, genes located within 10 kbp of significant SNPs were found through the Soybase database [31]. Many SNPs were found to be significant across locations and years. To limit the search, only SNPs that were found to be significant or close to significant in multiple environments were used.

### 4.4. Protein, Oil, and Sugar Content

To measure protein and oil content, near-infrared reflectance spectroscopy (NIRS) was performed. Each sample and replication was run twice on a DA7250 NIR Analyzer from Perten, and the protein and oil content were averaged for the sample. For each sample, about 60 mL was used. Only samples that had a yellow or green seed coat could be used. For both protein and oil content, 228 samples were used in both locations for 2019, 233 in both locations for 2020, 328 in BB 2021, and 258 in W 2021.

For sucrose, raffinose, and stachyose content, the protocol in Lord et al. 2011 was followed [32]. Briefly, 180 seed samples were first ground using a water-cooled grinder until the seeds became fine powder. All samples were weighed out to 0.1 g, and then 1.0 mL of HPLC-grade water was added to each sample. All samples were shaken for 15 min at 400 strokes per minute. Samples were centrifuged at 13.2 rpm for 15 min, and 0.5 mL of the supernatant was transferred to a new 2.0 mL centrifuge tube. Then, 0.7 mL of acetonitrile (ACN) was added, and all tubes were mixed by inverting multiple times. Samples sat for 1 h at room temperature and were then centrifuged at 17,000× *g* for 15 min. After centrifuging, 100 μL of the supernatant was mixed with 900 μL of 65% ACN and filtered through a 0.2 μm membrane into an HPLC sample vial. HPLC was used to determine sucrose content according to Lord et al. 2021. All samples were adjusted for moisture, and technical replicates were averaged together. Sugar analysis was only performed on 2019 samples due to there being no significance found between test weight and sugar. To determine correlation between high test weight and the three seed composition traits, protein, oil, and sugar content, R was used to calculate the Pearson’s correlation coefficient. An α of <0.05 was used to determine if the correlations were significant.

## Figures and Tables

**Figure 1 plants-12-02997-f001:**
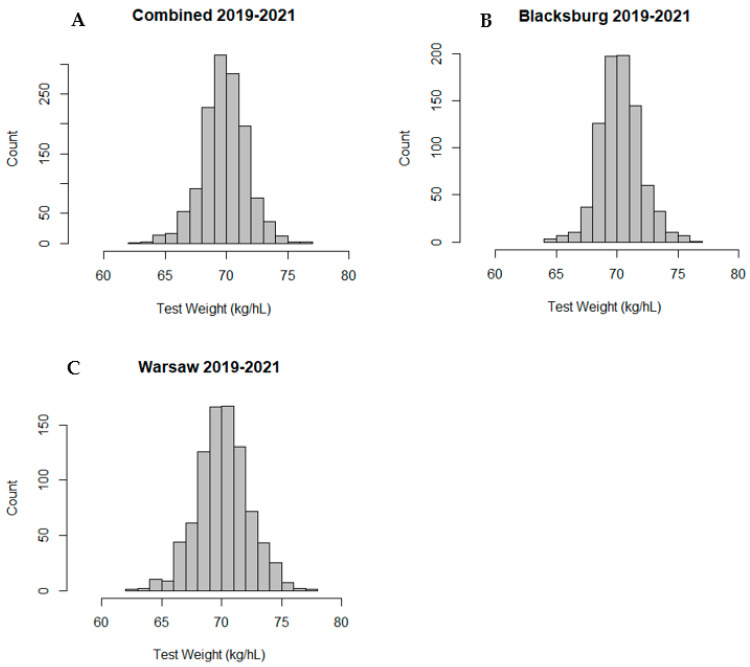
Frequencies of test weight values across both locations (**A**), Blacksburg (**B**), and Warsaw (**C**) for 2019–2021.

**Figure 2 plants-12-02997-f002:**
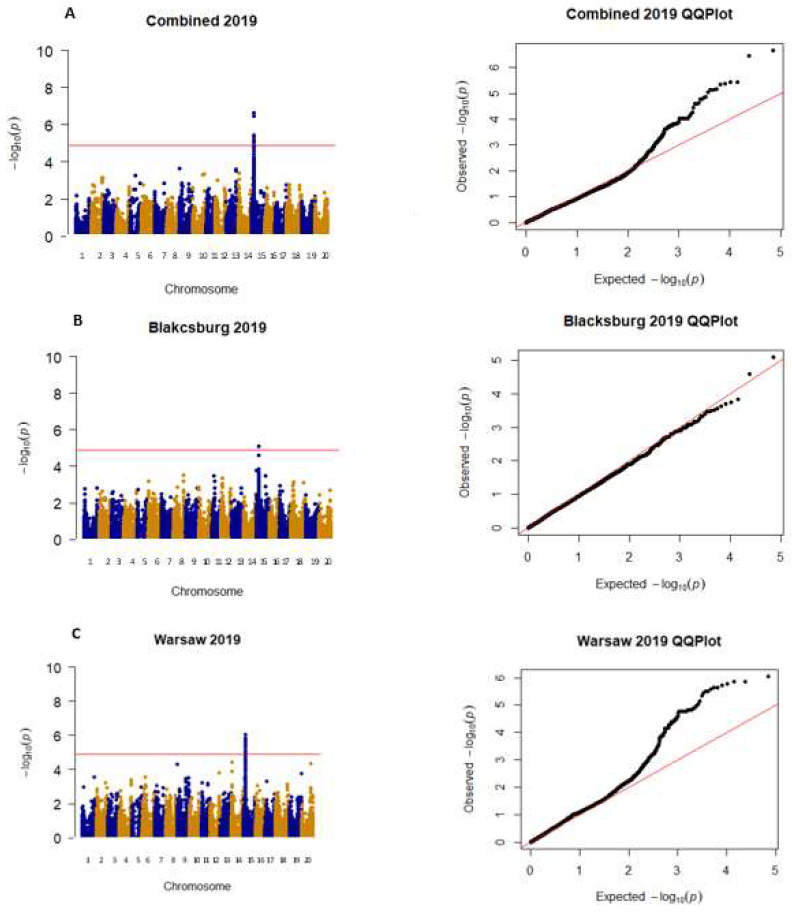
Manhattan and QQ plots for both locations (**A**), Blacksburg (**B**), and Warsaw (**C**) for 2019. Chromosomes in the Manhattan plots are shown in alternating colors, each dot represents one SNP, and the significance threshold is represented by the red line at −log10(p) of 4.9. QQ plots show observed −log10(p) plotted against expected −log10(p).

**Figure 3 plants-12-02997-f003:**
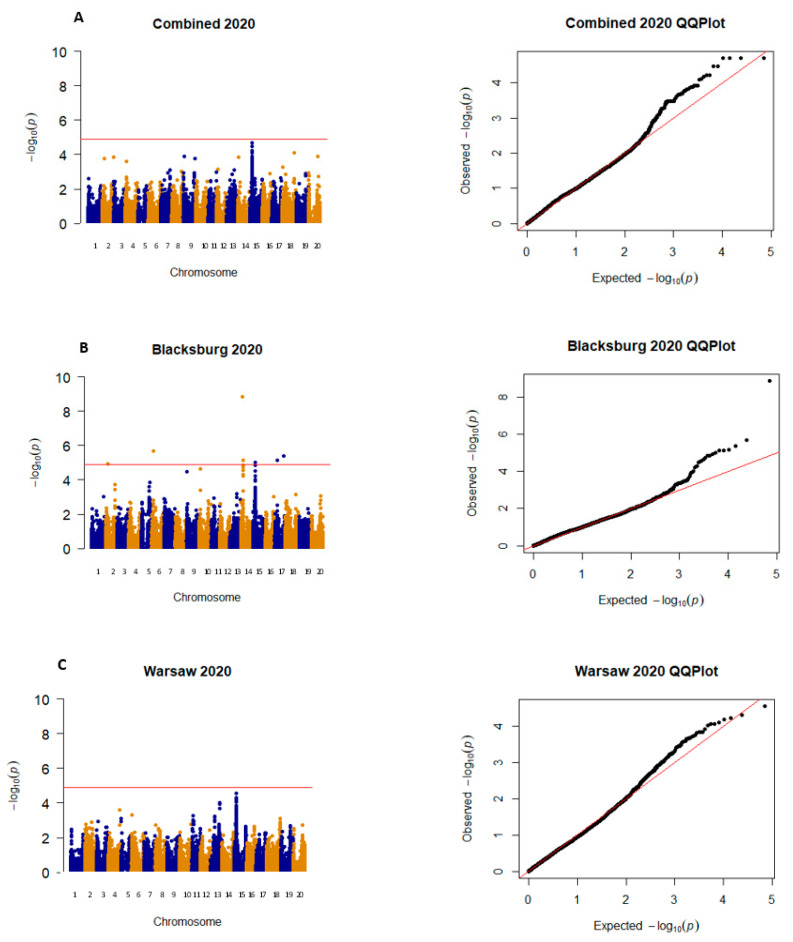
Manhattan and QQ plots for both locations (**A**), Blacksburg (**B**), and Warsaw (**C**) for 2020. Chromosomes in the Manhattan plots are shown in alternating colors, each dot represents one SNP, and the significance threshold is represented by the red line at −log10(p) of 4.9. QQ plots show observed −log10(p) plotted against expected −log10(p).

**Figure 4 plants-12-02997-f004:**
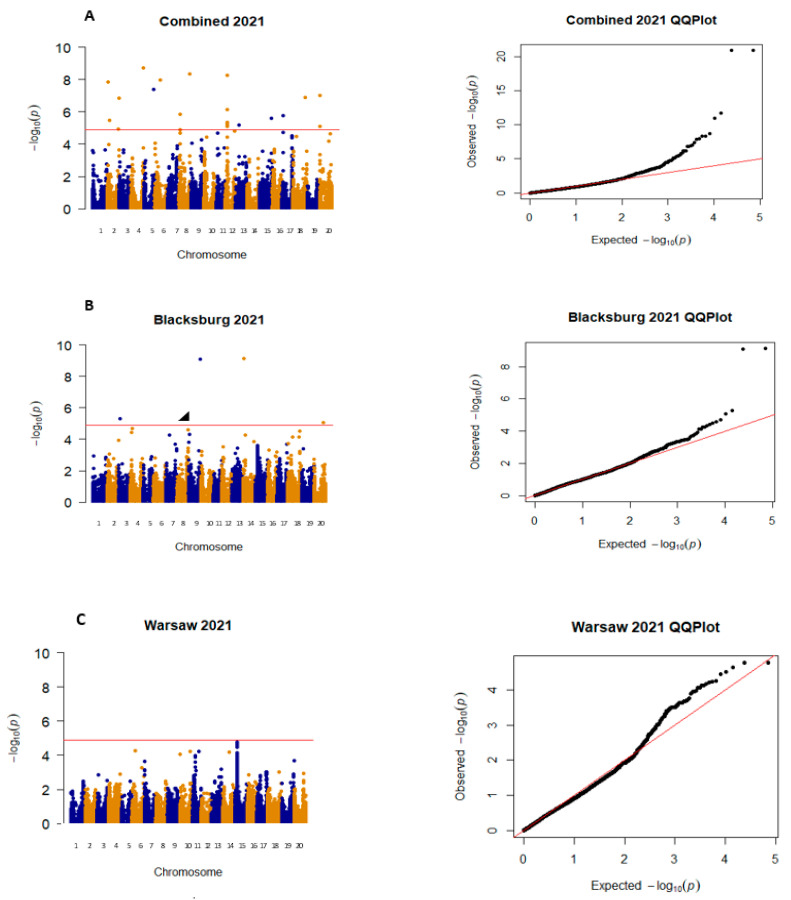
Manhattan and QQ plots for both locations (**A**), Blacksburg (**B**), and Warsaw (**C**) for 2021. Chromosomes in the Manhattan plots are shown in alternating colors, each dot represents one SNP, and the significance threshold is represented by the red line at −log10(p) of 4.9. QQ plots show observed −log10(p) plotted against expected −log10(p).

**Table 1 plants-12-02997-t001:** Significant SNPs on chromosomes 14 and 15 associated with test weight in soybean.

						Environment (−log10(p))					
Marker	Chr.	Position	BB 2019	BB 2020	BB 2021	W 2019	W 2020	W 2021	Combined 2019	Combined 2020	Combined 2021
ss715618482	14	3559612	NS ^b^	NS	NS	4.41 ^a^	NS	NS	NS	NS	NS
ss715619843	14	7207504	NS	8.86	NS	NS	NS	NS	NS	NS	NS
ss715618025	14	2201645	NS	NS	9.13	NS	NS	NS	NS	NS	NS
ss715623162	15	8758404	NS	5.37	NS	NS	NS	NS	NS	NS	NS
ss715623211	15	9205168	NS	NS	NS	5.51	NS	NS	5.13	NS	NS
ss715623224	15	9279044	NS	NS	NS	5.85	NS	NS	5.44	NS	NS
ss715620221	15	9383632	NS	NS	NS	5.72	NS	NS	5.33	NS	NS
ss715623250	15	9557248	NS	NS	NS	4.99	NS	NS	5.39	NS	NS
ss715623269	15	9748128	4.60 ^a^	NS	NS	5.76	NS	NS	NS	NS	NS
ss715623270	15	9749617	5.09	NS	NS	6.04	NS	NS	6.66	NS	NS
ss715623292	15	9927090	NS	NS	NS	NS	NS	4.77 ^a^	NS	4.70 ^a^	NS
ss715620172	15	10176737	NS	NS	NS	NS	4.56 ^a^	NS	NS	NS	NS

^a^ These values are just below the significance threshold. ^b^ NS = Not significant.

**Table 2 plants-12-02997-t002:** Summary of candidate genes.

Chromosome	SNP (Position)	Gene	Expression Pattern	Function
14	ss715618482 (3559612)	Glyma.14g046800	leaf, flower, pod, seed	Serine phosphatase
14	ss715619843 (7207504)	Glyma.14g082900	leaf, flower, pod	Cytochrome subfamily
14	ss715618025 (2201645)	Glyma.15g111700	leaf, flower, pod, seed,	Ribosomal protein
15	ss715623162 (8758404)	Glyma.14g030400	flower	Dioxygenase
15	ss715623211 (9205168)	Glyma.15g117100	leaf and pod	Transcriptional regulation
15	ss715623224 (9279044)	Glyma.15g118100	leaf, flower, pod, seed	Pentatricopeptide protein
15	ss715620221 (9383632)	Glyma.15g119200	pod, seed	Seed storage protein
15	ss715623269 (9748128)	Glyma.15g122800	root hair, root tip	RNA and protein binding
15	ss715623270 (9749617)	Glyma.15g122800	root hair, root tip	RNA and protein binding
15	ss715623292 (9927090)	Glyma.15g125000	root	Serves as a methyltransferase
15	ss715620172 (10176737)	Glyma.15g127900	root	Unknown

**Table 3 plants-12-02997-t003:** Correlation between test weight and five important seed composition traits.

Trait	BB 2019	W 2019	BB 2020	W 2020	BB 2021	W 2021
Protein	−0.115 *	−0.136 *	0.021	0.390 *	−0.052	0.047
Oil	−0.174 *	−0.297 *	−0.203 *	−0.265 *	−0.024	−0.387 *
Raffinose	−0.175 *	−0.118	NA	NA	NA	NA
Sucrose	−0.086	−0.08	NA	NA	NA	NA
Stachyose	−0.003	−0.063	NA	NA	NA	NA

* These correlations were found to be significant at α = 0.05.

## Data Availability

All data were generated at Virginia Tech facilities and can be provided upon request. Additionally, they are available on figshare.org accessible 26 July 2023 (https://figshare.com/articles/dataset/Test_Weight_Data_For_2019_to_2021_BB_and_W_Virginia_csv/23786286).

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
