# Peer review of "Determining Genetic Markers and Seed Compositions Related to High Test Weight in Glycine max"

_plants, 2023, doi:10.3390/plants12162997_

Round 1

Reviewer 1 Report

This manuscript documents observations of Soy50K SNPs in a population of soybean accessions (390) from the “Southern Core Collection”.  It identifies SNPs associated with the “Test Weight” phenotype of these accessions in multiple locations over multiple years.  GWAS analysis indicated numerous SNPs were significantly associated with this trait over the locations and years.  However, only a small minority of the SNPs (2) were significant across all environments and years.  Based on these SNPs the authors identified candidate genes

1.     Why did you choose to report “Test Weight” rather than the usual “Seed Weight”(weight of 100 seeds)?  It would be more comparable to other literature. If the seeds are still available, you should present both.

2.     Line 104 “Warsaw did not have any SNPs above LOD of xxx in 2020 and 2021”

3.     Lines 97 – 100. These sentences are very unclear.

4.     Lines 100 – 103  In line 100 you state the following “In 2019, all significant SNPs were found on chromosome 100 15” yet in lines 101-103 you state that many other significant SNPs were found in other chromosomes but not in all years/locations.  I think you should rearrange these sentences to start with lines 101 to 103 first then lines 100 -102.  While restricting your focus to the most “durable” SNPs does make sense, are you concerned that you are seeing only the “core” genes of “Test Weight” and not really the sensors that lead to the “core” functions?   Obviously, the core genes have variation segregating because you found SNPs but are those other loci important?  Could they be better breeding targets especially if the “core” genes are important to other phenotypes as well? Can you correlate environmental conditions at the locations (rain fall, soil type, GDD, insolation, max/min temp) with those “other” SNPs that you did not consider?

5.     Table 1. Title says that these are the data that supports your claim that the listed SNPs are “significant” across “multiple” environments.  This is not supported by the table. Only SNPs ss715623269 and ss715623270 are significant in “multiple” environments.

6.     Table 2. Based on the item 5 above, this table needs revision.

7.     The candidate gene list might be reduced if you used the expression patterns of those genes in various tissues such as seed or pod.  There are several “Gene Expression Atlases” accessible online for soybean that could easily be used to survey the candidate genes for the expected pattern of expression for a seed related trait.

8.     Discussion section needs revision based on item 5 above.

9.     The Test Weight data is not presented, only the summary data and not all of that.  There should be a supplemental table that contains the measurements from the reps of each of the 390 accessions along with their location data.  If a supplemental table of that size cannot be associated with this manuscript, the data must be deposited in a permanent repository like datadryad (datadryad.org) or figshare (figshare.org). Saying that the data is “available on request” in practice only lasts as long as the corresponding author’s listed email address or less.  Since these data were derived from the “Southern Core Collection” they may be of use to others in the future, thus they must be preserved in a permanent repository that practices the FAIR principles.

Reviewer 2 Report

The manuscript “Determining Genetic Markers and Seed Compositions Related to High Test Weight in Glycine max” identified that the genes responsible for test weight might be located on chromosome 15. Their findings offer valuable insights into the genetic factors influencing test weight and present promising applications for targeted breeding efforts.

The experiments were well done but I had some concerns regarding data representation and references.

·       The title says glycine max but the authors have not reported anywhere in the manuscript about the crop glycine max. They have just used soybean everywhere, maybe change the title or add a sentence in the introduction about the crop.

·       Figure 1 doesn’t have a statistical error bar. I would request the authors to add the P-value or show the statistical significance for the data sets.

·       The reviewer would like to suggest that the authors add more references to the discussion section.

Round 2

Reviewer 1 Report

It appears that the authors have responded to all my points.

However, I was unable to confirm the submission of your data to figshare.  Upon submission, you should have received a URL/URI to the data.  That URL should also accompany the statement "Additionally it is available on figshare.org (URL)"

Author Response

We apologize for not including the URL. We included the URL (https://figshare.com/articles/dataset/Test_Weight_Data_For_2019_to_2021_BB_and_W_Virginia_csv/23786286) in the data availability statement.